# Genome Analysis for Cholesterol-Lowing Action and Bacteriocin Production of *Lactiplantibacillus plantarum* WLPL21 and ZDY04 from Traditional Chinese Fermented Foods

**DOI:** 10.3390/microorganisms12010181

**Published:** 2024-01-17

**Authors:** Kui Zhao, Liang Qiu, Xueying Tao, Zhihong Zhang, Hua Wei

**Affiliations:** 1State Key Laboratory of Food Science and Resources, Nanchang University, Nanchang 330047, China; zk17770843514@163.com (K.Z.); taoxueying@ncu.edu.cn (X.T.); azhangzhihong@163.com (Z.Z.); 2Centre for Translational Medicine, Jiangxi University of Traditional Chinese Medicine, Nanchang 330006, China; qiuliang2016@126.com

**Keywords:** genome analysis, cholesterol, *Lactiplantibacillus plantarum* WLPL21, *Lactiplantibacillus plantarum* ZDY04, Chinese fermented foods

## Abstract

*Lactiplantibacillus plantarum*, a typical ecological species against pathogens, used due to its bacteriocin yield in fermented foods, was proven to have the capacity to lower cholesterol. In this study, using *L. plantarum* ATCC8014 as the control, *L. plantarum* WLPL21 and ZDY04 were probed with whole-genome sequencing to ascertain their potential ability to lower cholesterol and yield bacteriocins, as well as to further evaluate their survival capacity in vitro. Our results showed 386 transport-system genes in both *L. plantarum* WLPL21 and ZDY04. Correspondingly, the in vitro results showed that *L. plantarum* WLPL21 and ZDY04 could remove cholesterol at 49.23% and 41.97%, respectively, which is 1.89 and 1.61 times that of *L. plantarum* ATCC8014. The survival rates of *L. plantarum* WLPL21 and ZDY04 in 1% H_2_O_2_, pH 3.0, and 0.3% bile salt were higher than those of *L. plantarum* ATCC8014. Our results exhibited a complete gene cluster for bacteriocin production encoded by *L. plantarum* WLPL21 and ZDY04, including *pln*JKR, *pln*PQAB, *pln*EFI, *pln*SUVWY, and *pln*JK; and *pln*MN, *pln*PQA and *pln*EFI, respectively, compared with only *pln*EF in *L. plantarum* ATCC8014. The present study suggests that the combination of genomic analysis with in vitro evaluations might be useful for exploring the potential functions of probiotics.

## 1. Introduction

Traditional fermented foods (TFFs) have lasted for thousands of years, and they have attracted global attention over the last five years, especially in China and other Asian countries, e.g., Korea and Japan [1,2], due to their advantages of prolonging preservation time, improving the flavor of food, positively modulating gut microbiota, and being beneficial for health, e.g., alleviating DSS-induced colitis and ameliorating high-cholesterol-diet-induced hypercholesterolemia [3,4].

Plenty of lactic acid bacteria (LAB) participate in the fermentation processes of TFFs in China and other Asian countries [5,6]. LAB ecological (LABE) strains from different fermented food matrices might present different biological properties despite belonging to the same species. In fact, during the evolution process, LABE strains keep their key genes and specific gene clusters to adapt to pressure, survive, adhere, and multiply and colonize in different niches (e.g., food, animals, and human beings).

The genomes of LAB originally isolated from fermented foods have been sequenced in recent decades. Liu et al. sequenced the genome of *L. plantarum* strain 5-2 isolated from fermented soybean and conducted a comparative analysis with *L. plantarum* ST-III, JDM1, and WCFS1 [7]. Moreover, certain genome features, including gene annotation, phylogenic analysis, transport systems, bacteriocin production, and CRISPR structure, have been discussed across many publications. For example, the genome of *Levilactobacillus brevis* MYSN105, isolated from an Indian traditional fermented food, *Poza,* had 2926 CDSs and a 46.27% GC content [8]. Gene CRISPR clusters and bacteriocins of *L. plantarum* LT52 isolated from raw-milk cheese were also analyzed [9].

Cholesterol is an essential part of membranes and thus participates in vital movement; however, excessive cholesterol intake will increase the risk of metabolic syndromes. In the last decade, curbing cholesterol intake through the administration of LABE supplements has come to be considered a valid measure to lower the risk of hypercholesterolemia and even cardiovascular disease [10,11]. For instance, Zhai et al. showed that *L. rhamnosus* GG protected against atherosclerosis by improving ketone body synthesis [12]. At that time, there was less information than there is now on the key genes and gene clusters of LABE strains in charge of lowering cholesterol, especially for *L. plantarum* from TFFs in China.

Besides their cholesterol-lowering ability, their antagonistic capacity against pathogens is one of the important indexes for strengthening the probiotic properties of LABE strains. This capacity is mostly related to the production of organic acids and partially related to the yield of bacteriocin, one of the key secondary metabolites for partial LABE strains [13,14]. As a common species in TFFs, *Lactobacillus* spp.’s gene cluster and yield amount of bacteriocins depend on different niches. For instance, Liu et al. reported that *L. paraplantarum* L-ZS9 from fermented sausage could produce class Ⅱ bacteriocins in a whole-genome analysis [15]. Goel et al. found that *L. plantarum* DHCU70 from fermented milk dahi and *L. plantarum* DKP1 from kinema (fermented soybean, India) contained a 20.5 kb long gene region consisting of 23 genes (pln) [16]. Upendra et al. showed that the bacteriocin yield of *Pediococcus pentosaceus* reached 2.4 mg/L after fermentation optimization [17].

In our previous study, *L. plantarum* WLPL21 and ZDY04 from Chinese fermented soybeans and pickled vegetables, respectively, were shown to play a great role in the alleviation of hypercholesterolemia in a high-cholesterol-diet-induced model in vivo [18]. However, elucidating its cholesterol-lowering mechanism and the existence of bacteriocin genes still stand as research hurdles. To make progress forward, in this study, we performed a comparative genomic analysis of *L. plantarum* WLPL21 and ZDY04 in consideration of their gene functions and categories, phylogenetics, transport genes, and bacteriocins. We also validated the findings in vitro by using *L. plantarum* ATCC8014 as a control, with the aim of probing the potential molecular mechanisms of *L. plantarum* in lowering cholesterol.

## 2. Materials and Methods

### 2.1. Bacterial Strain and Culture Conditions

*L. plantarum* WLPL21 and *L. plantarum* ZDY04, isolated from Chinese fermented soybeans and pickled vegetables [18], were cultured under anaerobic conditions at 37 °C in sterile deManRogosa Sharpe (MRS broth; Beijing Solarbio Science & Technology Co., Ltd., Beijing, China) in an anaerobic incubator (GeneScience Anaerobox IV, San Francisco, CA, USA) for 12 h.

### 2.2. Genome Sequencing and Annotation

Genomic DNA of *L. plantarum* WLPL21 and ZDY04 was extracted and purified based on the method of Surachat et al., as described previously, with minor changes [19]. DNA extraction was performed with a DNeasy extraction kit (QIAGEN, Hilden, Germany) following the manufacturer’s instructions. Briefly, the bacterial cell pellet (10^9^ cfu) was resuspended in 180 μL of an enzymatic lysis buffer and incubated at 37 °C for 30 min. Next, 25 μL of proteinase K and 200 μL of Buffer AL were added and mixed before incubation at 56 °C for 30 min. Then, 200 μL of ethanol was added to the DNA sample and it was centrifuged with the DNeasy Mini spin column at 610 g for 1 min. After that, the DNA was washed with 500 μL of Buffer AW2 and then eluted with buffer AE. Finally, the DNA concentration in the eluate was measured using a spectrophotometer at 260 nm. The ratio of the absorbance at 260 nm and 280 nm (A260/A280) provided an estimate of the purity of the DNA via agarose gel electrophoresis.

Then, the DNA concentration, quality, and integrity were determined by using a Qubit Fluorometer (Invitrogen, Waltham, MA, USA) and a NanoDrop Spectrophotometer (Thermo, Waltham, MA, USA). Sequencing libraries were generated using the TruSeq DNA Sample Preparation Kit (Illumina, San Diego, CA, USA) and the Template Prep Kit (Pacific Biosciences, Menlo Park, CA, USA). Genome sequencing was then performed at Personal Biotechnology Company (Shanghai, China) by using the platform of Pacific Biosciences and the Illumina Miseq.

Data assembly proceeded after adapter contamination and data filtering with AdapterRemoval and SOAPec [20,21]. The filtered reads were assembled with SPAdes and A5-miseq to construct scaffolds and contigs [22,23]. Canu software (V-1.6) was used to assemble the data obtained through Pacbio platform sequencing [24]. Subsequently, all assembled results were integrated to generate a complete sequence. Finally, genome sequence was performed after rectification by using Pilon software (V-1.4) [25].

Genome function elements’ prediction included the prediction of a coding gene, non-coding RNA, and clustered regularly interspaced short palindromic repeats (CRISPRs). Gene prediction was performed with Glimmer 3.02 [26]. tRNAscan-SE, RNAmmer, and Rfam were used to find tRNA, rRNA, and other ncRNA, respectively. CRISPRs were identified with the CRISPR recognition tool [27,28,29,30].

Function annotation was completed through a BLAST search against different databases, including the Non-Redundant Protein Database (NR), Gene Ontology (GO), Kyoto Encyclopedia of Gene and Genomes (KEGG), Cluster of Orthologous Groups of proteins (COG), and SWISS-PROT [31,32,33,34]. CGview was used to give an overview of the genome information [35].

### 2.3. Phylogenetic Analysis

The complete sequence of *L. plantarum* ATCC8014 (Accession: NC_019916.1) was obtained from the National Center for Biotechnology Information (NCBI) database. The 16S rRNA and *bsh* genes were available in the NCBI database. We performed multiple sequence alignment of 16S rRNA and *bsh* genes in the genome of *L. plantarum* WLPL21 (Accession: NZ_CP122378.1), ZDY04 (Accession: NZ_CP122406.1), and ATCC8014, and other related genes were aligned; then, a maximum likelihood (ML) phylogenetic tree was constructed using MEGA 7 software. Genome collinearity of *L. plantarum* WLPL21, ZDY04, and ATCC8014 was analyzed using Mauve 20150226.

### 2.4. Identification of Bacteriocin-Encoding Genes

Different plantaricin (*pln*) genes were identified in the genome of *L. plantarum* WLPL21 and ZDY04. In addition, those in *L. plantarum* ATCC8014 were gained from the NCBI database. Then, a gene cluster was obtained from www.chiplot.online (accessed on 6 December 2022).

### 2.5. CRISPR Analysis

The genes relevant to CRISPR in *L. plantarum* ATCC8014 were not found in the NCBI database. Three CRISPR structures of *L. plantarum* WLPL21 and ZDY04 and one gene cluster were found based on genome analysis. Then, a gene cluster was obtained from www.chiplot.online (accessed on 9 March 2023).

### 2.6. Probiotic Characteristics

To evaluate antioxidant activity and ability to tolerate acid and bile salts, MRS broth modified with 1% H_2_O_2_, 0.3% bile salt, and pH 3.0 with 1 M HCl was prepared. *L. plantarum* WLPL21, ZDY04, and ATCC8014 were cultured under anaerobic conditions at 37 °C for 12 h in MRS broth. The bacterial solution was centrifuged 3 times and then resuspended with PBS at the concentration of 10^9^ cfu/mL. After that, the bacterial solutions of three probiotics were added separately into three modified MRS broths at the proportion of 1%. After 2 h incubation, the viable bacteria were counted according to plate counts by using a tenfold dilution method with 3 replications.

A cholesterol removal assay was designed based on Xu et al. [36] with minor changes. MRS-Thio-Ox-CHOL medium was created as MRS medium containing 0.3% bovine bile salt, 0.2% sodium mercaptoacetate, and 0.1 mg/mL cholesterol solution. LAB was cultured under anaerobic conditions at 37 °C for 12 h, and the supernatant was harvested through centrifugation at 4 °C and 12,000 rpm for 10 min. The bacterial solution was resuspended in MRS-THIO broth containing 0.3% oxgall and 0.2 mg/mL lysozyme and incubated at 37 °C for 15 min, and then the suspension was crashed using the supersonic method. The fragmented cell solution was maintained by supplementing MRS-THIO broth with 0.3% oxgall to the original volume and centrifuged at 4 °C and 12,000 rpm for 10 min. Cholesterol concentration in the fermentation supernatant, cell efflux solution, and cell fragment suspension was determined through the colorimetric method, in which MRS-THIO broth with 0.3% oxgall was set as the standard.

### 2.7. Statistical Analysis

The data and their normal distribution (p of Shapiro–Wilk normality greater than 0.05 was thought to conform to normal distribution) were analyzed using GraphPad Prism 7 statistical software. The results were expressed as the mean ± standard deviation (SD). All the data that passed the verification of normal distribution were analyzed via one-way analysis of variance and Tukey’s multiple comparisons tests, which were used for comparisons between groups, while the others were analyzed via a non-parametric test. *p*-values less than 0.05 were considered statistically significant.

## 3. Results

### 3.1. Genome Analysis of L. plantarum

The complete genome of *L. plantarum* WLPL21 and ZDY04 contained a single circular chromosome of 3,304,253 and 3,304,238 bp with no plasmids (Figure 1). The GC content of the two strains was 44.56% and that of *L. plantarum* ATCC8014 was 44.43%. The open reading frame (ORF) is the theoretical coding region of amino acids, and 3072 and 3075 genes, which represented 83.54% and 83.56% of the total length, were predicted as the ORF, respectively, in *L. plantarum* WLPL21 and ZDY04 (Table 1). They both contained 61 ncRNA, 16 rRNA, and 65 tRNA, corresponding to a cyl-alkaline amino acid (Sup) and all 20 natural amino acids: Leu (6 sequences); Gly and Arg (5); Met, Thr, Lys, Asp (only in *L. plantarum* ZDY04), and Ser (4); Gln, Asn, Ile, Asp, and Glu (only in *L. plantarum* WLPL21), and Pro, Val, and Ala (3); Glu (only in *L. plantarum* ZDY04), Tyr, His, and Phe (2); Cys, Trp, and Sup (1). Clustered Regularly Interspaced Short Palindromic Repeats (CRISPRs) have usually been used for genetic operations over the past decades, and when applying those, two strains were identified to have three repeat regions.

There were 3072 and 3075 genes, besides 451 genes and 455 genes not in eggnog (Figure 2), respectively, in *L. plantarum* WLPL21 and ZDY04, which were specifically assigned to clusters of COG families into 20 functional categories. Except for 640 and 641 unknown genes in *L. plantarum* WLPL21 and ZDY04, respectively, the other genes include energy production and conversion (119 genes); cell cycle control, cell division, and chromosome partitioning (25 genes); amino acid transport and metabolism (203 genes); nucleotide transport and metabolism (84 genes); carbohydrate transport and metabolism (261 genes); coenzyme transport and metabolism (75 genes); lipid transport and metabolism (55 genes); translation, ribosomal structure, and biogenesis (149 genes); transcription (262 genes); replication, recombination, and repair (159 genes); cell wall/membrane/envelope biogenesis (192 genes); posttranslational modification, protein turnover, and chaperones (64 genes); inorganic ion transport and metabolism (145 genes); secondary metabolites’ biosynthesis, transport, and catabolism (21 genes); signal transduction mechanisms (84 genes); intracellular trafficking, secretion, and vesicular transport (21 genes); defense mechanisms (60 genes); and extracellular structures (1 gene).

The KEGG (Kyoto Encyclopedia of Genes and Genomes) ortholog aims to annotate the relevant information of molecular networks across different species. GO (Gene Ontology), meanwhile, has three classifications: molecular function, cellular component, and biological process. The results of applying KEGG and GO to *L. plantarum* WLPL21 and ZDY04 are presented in Figure 3 and Figure 4. Our results show that there were no differences in KEGG and GO between the genomes of *L. plantarum* WLPL21 and ZDY04.

### 3.2. Phylogenetic Analysis

The phylogenetic tree is used to describe the relationships among different species. Based on 16S rRNA genes, *L. plantarum* WLPL21 and ZDY04 showed 99% similarity with other *L. plantarum* but less than 85% with other *Lactobacillus* spp., suggesting that there were greater similarities in *L. plantarum* than another *Lactobacillus* spp. (Figure 5). Therefore, our results showed that *L. plantarum* WLPL21, ZDY04, and ATCC8014 had extremely close relations, but relatively distant relations with *L. paracasei*, *L. rhamnosus*, *L. fermentum*, *L. reuteri*, *L. delbrueckii*, *L. helveticus*, and *L. acidophilus*.

Bile salt hydrolase (BSH) is a kind of metabolic enzyme produced by probiotics in the process of growth and reproduction, which hydrolyzes combined bile salts and thus reduces the content of cholesterol in serum. In our previous study, *L. plantarum* WLPL21 and ZDY04 showed a great capacity to decrease cholesterol in vivo. Therefore, a phylogenetic tree based on the *bsh* sequence was used to calculate the capacity of decreasing cholesterol in *Lactobacillus* spp. *L. plantarum* WLPL21, ZDY04, ATCC8014, and ZDY2013 showed 100% similarity, while they showed less than 65% similarity with other *Lactobacillus* spp. (Figure 6). Therefore, *L. plantarum* WLPL21, ZDY04, and ATCC8014 may have similar bile acid hydrolase activity.

Genome collinearity analysis was used to compare the structure and function of the genome, and the relationships among different strains. Our results showed that there was a higher linear correlation among *L. plantarum* WLPL21, ZDY04, and ATCC 8014 (Figure 7). However, the gene rearrangement regions and different genome lengths indicated that *L. plantarum* WLPL21 and ZDY04 had a closer relationship than *L. plantarum* ATCC8014. We propose that subtle differences in genome distribution, such as those found here, may be related to the sources of strains.

### 3.3. Transport System

Reduction of cholesterol levels by probiotics relies not only on bile salt hydrolase but also on cholesterol transportation. In our results, 386, 386, and 232 genes for the transport system were identified in the genome of *L. plantarum* WLPL21, ZDY04, and ATCC 8014, among which ABC transport system and phosphotransferase system (PTS) genes were analyzed (Table 2). The ABC transporter is the transporter ATPase on the membrane, with the capacity for transporting inorganic ions, monosaccharides, glycans, cholesterol, phospholipids, amino acids, peptides, proteins, toxins, drugs, and antibiotics. ABC transporter genes in *L. plantarum* WLPL21 and ZDY04 included 1 betaine/proline/choline family ABC transporter ATP-binding protein gene, 19 peptide ABC transporter genes, 6 energy-coupling factor ABC transporter genes, 9 sugar ABC transporter genes, 71 ABC transporter ATP-binding protein genes, 24 amino acid ABC transporter genes, etc. Meanwhile, PTS is used to transport sugars and their derivatives via phosphorylation. We found 1 betaine/proline/choline family ABC transporter ATP-binding protein gene, 7 peptide ABC transporter genes, 4 energy-coupling factor ABC transporter genes, 4 sugar ABC transporter genes, 49 ABC transporter ATP-binding protein genes, 13 amino acid ABC transporter genes, etc., in *L. plantarum* ATCC8014. Our results showed that the functions of PTS genes included the transportation of glucose, sucrose, mannitol, cellobiose, mannose, fructose, sorbose, lactose, glucitol, sorbitol, N-acetylglucosamine, beta-glucoside, ascorbate, and galactitol. As such, analysis of the transport system, especially the ABC transport system and PTS, revealed that *L. plantarum* WLPL21 and ZDY04 might possess improved transport activity compared with *L. plantarum* ATCC8014.

### 3.4. Bacteriocin Production

Bacteriocins are a kind of peptide or leader peptide with antibacterial activity produced via ribosome synthesis during some bacteria’s metabolism. Because of their strong antimicrobial activity, they are beneficial to the colonization of bacteria in the gastrointestinal system and the recovery stability of the microbiota. In *L. plantarum* ATCC8014, the bacteriocin gene cluster was simple in that it only possessed the plantaricin-encoding genes *plnE* and *plnF*. In *L. plantarum* ZDY04, the plantaricin-encoding genes of *plnA*, *plnE*, *plnF*, *plnI*, *plnJ*, *plnK*, *plnM*, *plnN*, *plnP,* and *plnQ* formed its complete bacteriocin gene cluster. In addition, *L. plantarum* WLPL21 contained *plnA*, *plnE*, *plnF*, *plnI*, *plnJ*, *plnM*, *plnN*, *plnP*, *plnQ*, *plnS*, *plnU*, *plnV*, *plnW,* and *plnY* (Table 3). Besides some plantaricin genes, the response regulator transcription factor, export ABC transporter, bacteriocin ABC transport, regulator transcriptional factor, and ABC-type bacteriocin were also found in *L. plantarum* (Figure 8). Therefore, our results showed that *L. plantarum* WLPL21 had the most complex and comprehensive bacteriocin system among the three analyzed LABs.

### 3.5. CRISPR Analysis

CRISPR (Clustered Regularly Interspaced Short Palindromic Repeats) of the microbiology immune system has been used to edit genes in order to achieve molecular operations in the past decades. *L. plantarum* WLPL21 and ZDY04 had the same CRISPR system, but it was not found in *L. plantarum* ATCC8014 (Figure 9). We found 1026 repeat regions in the CRISPR gene cluster type Ⅱ CRISPR RNA-guided endonuclease Cas 9, type Ⅱ CRISPR-associated endonuclease Cas1, CRISPR-associated protein Cas 2, and type Ⅱ-A CRISPR-associated protein Csn 2. Our results showed that *L. plantarum* WLPL21 and ZDY04 were potential engineering probiotics due to their complete type Ⅱ-A CRISPR system.

### 3.6. Probiotic Characteristics In Vitro

To evaluate antioxidant activity, the ability of probiotics to tolerate 1% H_2_O_2_ was measured; therein, the survival rates of *L. plantarum* WLPL21 and ZDY04 were 113.70% and 105.50% compared to that of 65.58% in *L. plantarum* ATCC8014. The tolerance of acid and bile salt was also a key aspect in calculating the colonization of probiotics. *L. plantarum* WLPL21, ZDY04, and ATCC8014 all had great viability with survival of about 59.57%, 64.78%, and 51.85% and 66.97%, 60.01%, and 50.02%, respectively, in MRS broth with pH 3.0 and 0.3% bile salt (Figure 10A–C). A cholesterol removal test was used to evaluate the capacity of cholesterol-lowing probiotics in vitro. *L. plantarum* WLPL21 and ZDY04 could remove cholesterol at 49.23% and 41.97%, and it was 25.60% in *L. plantarum* ATCC8014 (Figure 10D). Therefore, *L. plantarum* WLPL21 and ZDY04 had excellent probiotic characteristics including resistance to acid and bile salt, and they had antioxidant cholesterol-lowering ability in vitro.

## 4. Discussion

*L. plantarum* presents a variety of biological characteristics (e.g., reducing cholesterol, antibacterial ability, and colonization), due to its long-term evolution in plenty of matrices in humans, animals, fermented food, etc. Therefore, genome comparison of the strains of *L. plantarum* will contribute to our understanding of the underlying mechanism of how probiotics survive in GI or fermented food and play a role in ameliorating host health. In this study, systematic genome analysis and potential probiotic characteristics of *L. plantarum* WLPL21 and ZDY04 were examined through bioinformatic analysis and in vitro testing (antioxidant activity, tolerance of acid and bile salt, and cholesterol removal ability), respectively, to identify the key genes and gene clusters in charge of cholesterol-lowing and antagonistic properties of bacteriocins.

A phylogenetic tree based on 16S rRNA was used to evaluate the affinities among different *Lactobacillus* spp.; therein, *L. plantarum* WLPL21 and ZDY04 were extremely similar to the other *L. plantarum* strains but showed little similarity with other *Lactobacillus* strains. Correspondingly, *L. delbrueckii*, *Weizmannia coagulans,* and *L. mudanjiangensis,* respectively, showed high similarities with the same species [37,38,39]. As a consequence of their similarity, *L. plantarum* WLPL21 and ZDY04 had plenty of similar potential probiotic functions. Furthermore, bile salt hydrolase plays a great role in the metabolism of cholesterol, and the results of phylogenetic analysis of *L. plantarum* WLPL21 and ZDY04 revealed that there were differences among *L. plantarum* strains and huge distances from the other LABs. Therefore, the genome of *L. plantarum* WLPL21 and ZDY04 had lots of similarities with the other *L. plantarum* stains, but some characteristics, especially in bile salt hydrolase, were different.

Genome collinearity analysis can reveal evolutionary changes in the nucleotides by aligning homologous regions of sequences. Surachat et al. reported that *Pediococcus acidilactici* HN9 had different gene regions compared to *P. acidilactici* JQII-5, SRCM100313, SRCM101189, and SRCM100424 [40]. The rearrangement region in *L. plantarum* ATCC8014 implied that *L. plantarum* WLPL21 and ZDY04 had a closer genetic relationship.

The ABCA and ABCG subfamilies belong to the ABC transporters, playing a great role in cholesterol transport both in vitro and in vivo [41]. Our results showed that the betaine/proline/choline family ABC transporter ATP-binding protein of *L. plantarum* WLPL21 and ZDY04 was closely related to cholesterol transportation and other ABC transport proteins. In contrast, *L. plantarum* ZLJ010 harbored 170 genes encoding the ABC transporter system components, most of which transported peptides, amino acids, and inorganic ions [42]. Moreover, encoding a lot of PTS, as we found was the case in the genome of *L. plantarum* WLPL21 and ZDY04, has a certain connection with probiotic properties [43]. Thus, genes of the ABC transport system and PTS might influence probiotic characteristics as well as cholesterol transportation.

Bacteriocins from LAB strains are considered to have high fitness due to their strong colonization activity in gut microbiota [44]. Four bacteriocin operons, *pln*JKR, *pln*PQAB, *pln*EFI, and *pln*SUVWY, were found in *L. plantarum* WLPL21, and another four bacteriocin operons (*pln*JK, *pln*MN, *pln*PQA, and *pln*EFI) in *L. plantarum* ZDY04; however, only *pln*EF was found in *L. plantarum* ATCC8014. It was reported that *L. plantarum* M4L1 encoded *pln*E, *pln*F, and *pln*K according to the whole-genome sequencing [45]. Consequently, we can surmise that *L. plantarum* WLPL21 and ZDY04 had relatively comprehensive bacteriocin gene clusters, which might enable them to establish a competitive advantage in GI, to help the host modulate the gut microbiota and even produce much more BSH for the removal of cholesterol.

Lastly, the tolerance of acid and bile salts and the antioxidant activity are important standards for use to evaluate the probiotic characteristics of LABs [46]. In our results, *L. plantarum* WLPL21 and ZDY04 presented significantly stronger antioxidant activity against 1% H_2_O_2_ than *L. plantarum* ATCC8014, and for tolerance of acid and bile salts, both strains also showed better properties. Consequently, we propose that an excellent antioxidant capacity of *L. plantarum* WLPL21 and ZDY04 may stand as a reason for their appearance in Chinese traditional fermented foods. As for the in vitro cholesterol removal test, both *L. plantarum* WLPL21 and ZDY04 presented more than 40% cholesterol removal capacity. Similarly, *L. plantarum* YS5 and ATCC14917 showed moderate cholesterol-removal capacities of 41.14% and 29% [47]. Based on their excellent in vitro activity, *L. plantarum* WLPL21 and ZDY04 could be regarded as potential probiotics, especially in lowering cholesterol.

## 5. Conclusions

Whole-genome analysis (genome basic information, phylogenetic analysis, transport system, bacteriocin, and CRISPR) and probiotic characteristics (antioxidant activity, tolerance of acid and bile salt, and cholesterol removal capacity) of *L. plantarum* WLPL21 and ZDY04 isolated from Chinese traditional fermented foods were explored in this study. The results showed that their excellent capacity for lowering cholesterol in vivo may be associated with their probiotic characteristics in vitro and encoding *bsh* genes; moreover, encoding genes for the transport system and bacteriocin production could accelerate cholesterol degradation in vivo. To extend these findings, further studies applying methods from molecular biology on the strains *L. plantarum* WLPL21 and ZDY04 are necessary to probe potential mechanisms of relevant genes for the removal of cholesterol in vivo.

## Figures and Tables

**Figure 1 microorganisms-12-00181-f001:**
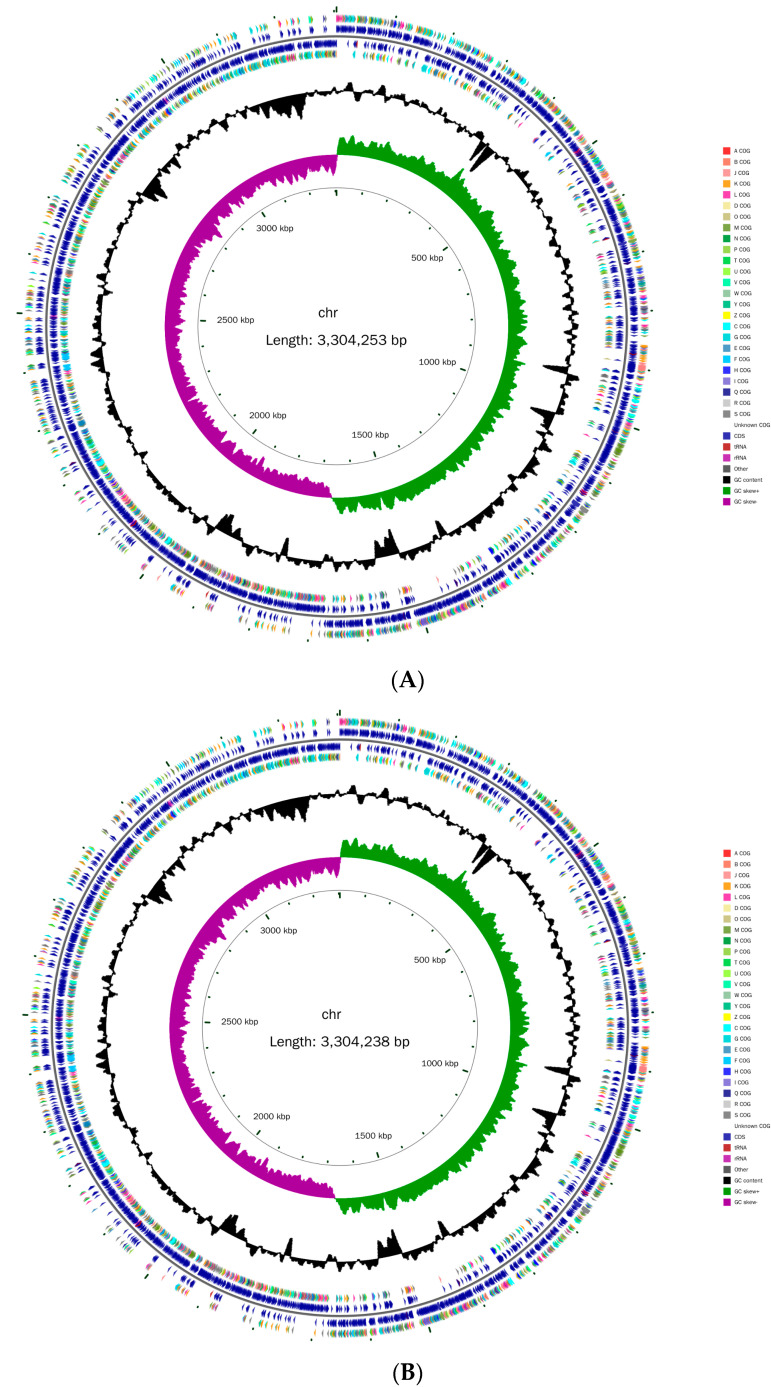
Circular genome map of *L. plantarum* WLPL21 (**A**) and ZDY04 (**B**). Marked information is displayed from the outer circle to innermost, as follows: the first circle represents the scale; the second circle represents the GC sketch; the third circle represents the GC content; the fourth and seventh circles represent the COG to which each CDS belongs; the fifth and sixth circles represent the positions of CDS, tRNA, and rRNA on the genome.

**Figure 2 microorganisms-12-00181-f002:**
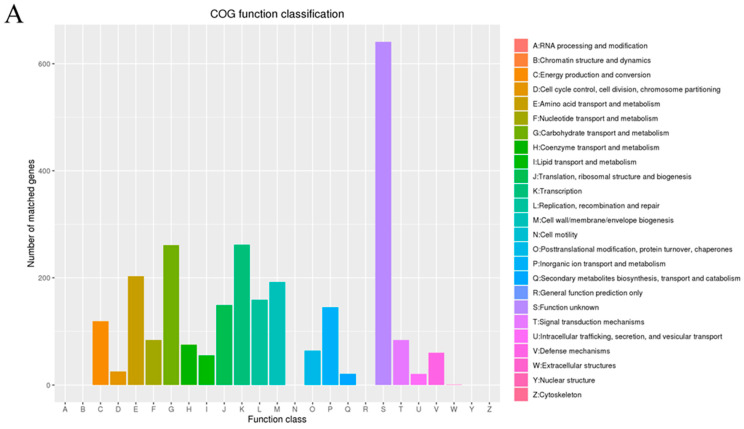
Function classification in COG (Cluster of Orthologous Groups of proteins) of *L. plantarum* WLPL21 (**A**) and ZDY04 (**B**).

**Figure 3 microorganisms-12-00181-f003:**
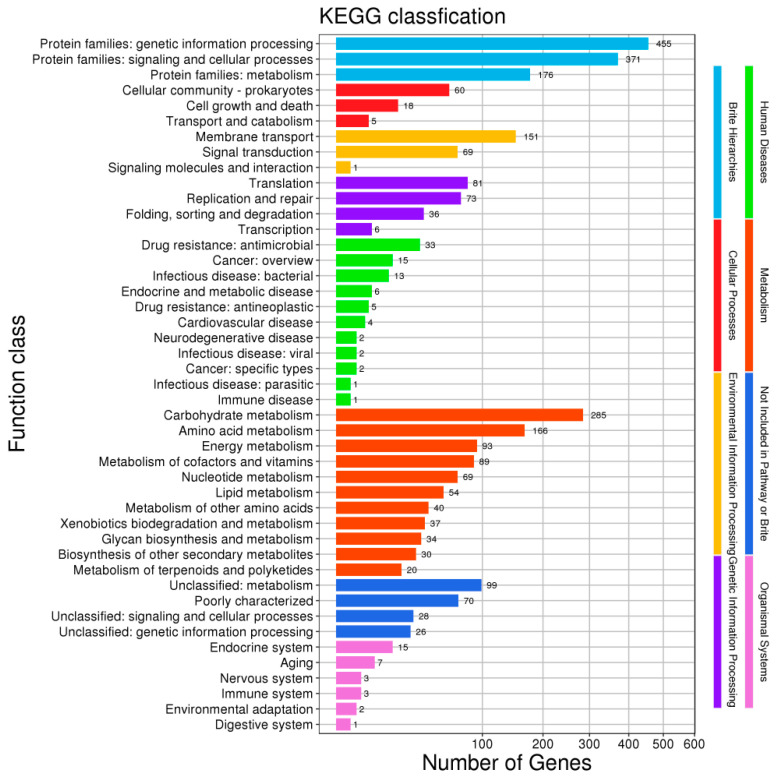
KEGG (Kyoto Encyclopedia of Gene and Genomes) classification of *L. plantarum* WLPL21 and ZDY04, which had the same KEGG classification.

**Figure 4 microorganisms-12-00181-f004:**
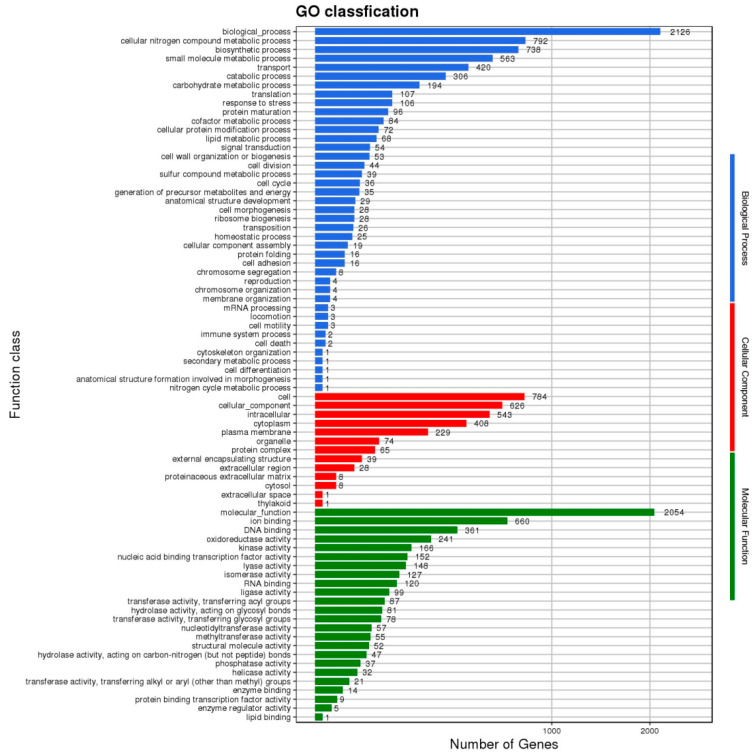
GO (Gene Ontology) classification of *L. plantarum* WLPL21 and ZDY04, which had the same GO classification.

**Figure 5 microorganisms-12-00181-f005:**
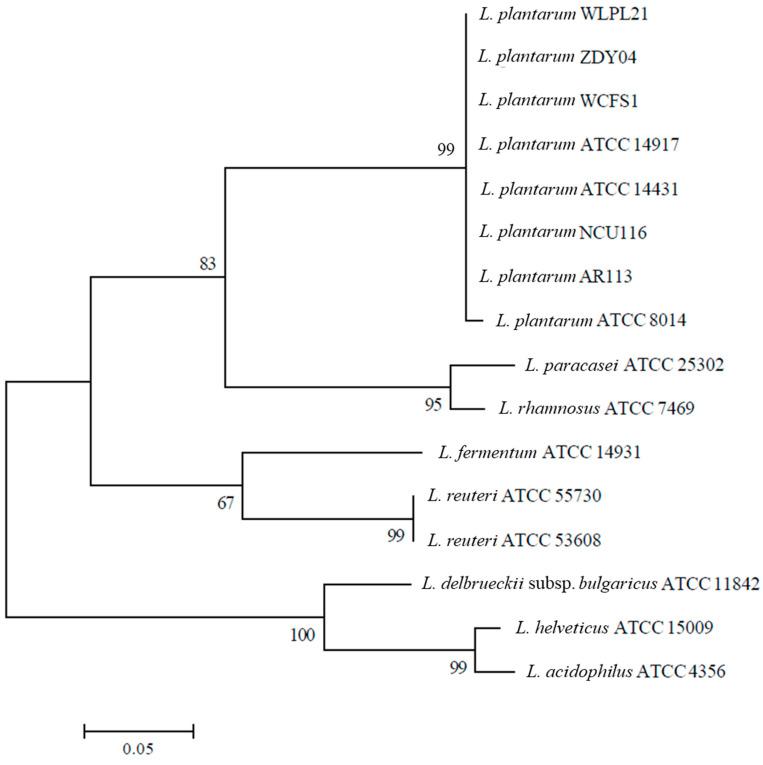
Construction of phylogenetic tree using NCBI database and the maximum likelihood estimate method based on 16S rRNA. The percent numbers at the nodes indicate the levels of bootstrap support based on 1000 replications. Bar, 0.05 nucleotide substitutions per site.

**Figure 6 microorganisms-12-00181-f006:**
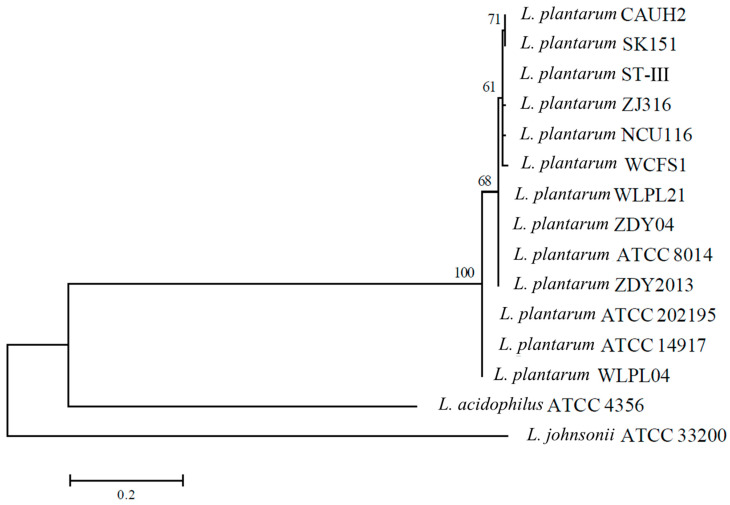
Construction of phylogenetic tree using NCBI database and the maximum likelihood estimate method based on bile salt hydrolase (*bsh*). The percent numbers at the nodes indicate the levels of bootstrap support based on 1000 replications. Bar, 0.2 nucleotide substitutions per site.

**Figure 7 microorganisms-12-00181-f007:**
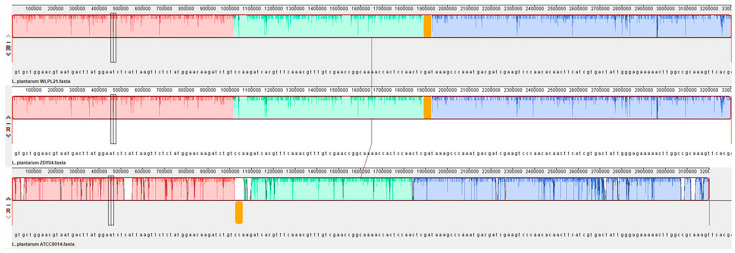
Genome collinearity analysis of *L. plantarum* WLPL21, ZDY04, and ATCC8014. Yellow area is gene rearrangement region. Red, green, and blue areas represent corresponding gene regions.

**Figure 8 microorganisms-12-00181-f008:**
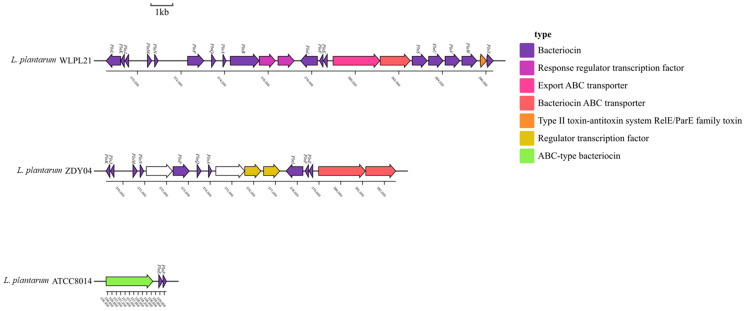
Bacteriocin cluster of *L. plantarum* WLPL21, ZDY04, and ATCC8014. Colored arrows represent bacteriocin-related genes, white arrows represent genes unrelated to bacteriocins.

**Figure 9 microorganisms-12-00181-f009:**
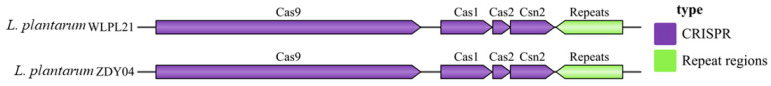
CRISPR (Clustered Regularly Interspaced Short Palindromic Repeats) cluster in *L. plantarum* WLPL21 and ZDY04 but not in ATCC8014.

**Figure 10 microorganisms-12-00181-f010:**
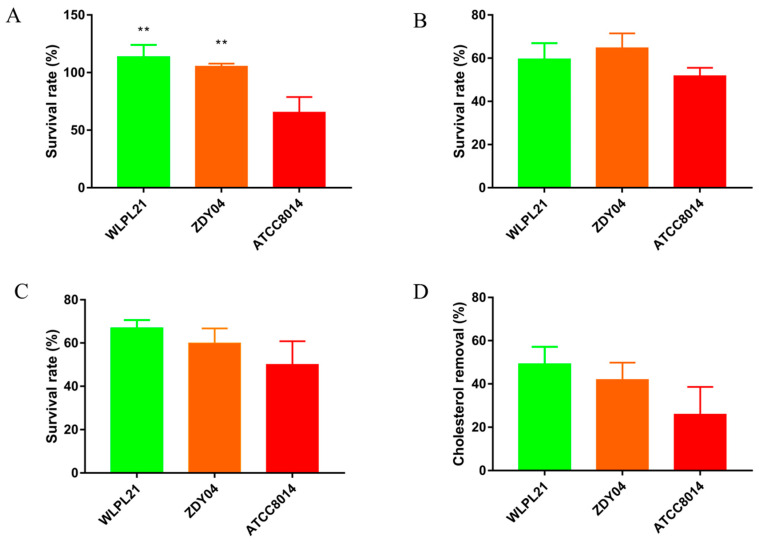
Survival rates of cells under conditions of 1% H_2_O_2_ (**A**), pH 3.0 (**B**), and 0.3% bile salt (**C**), and cholesterol removal rates (**D**) of *L. plantarum* WLPL21, ZDY04, and ATCC8014. ** *p* < 0.01, data in WLPL21 and ZDY04 were compared to ATCC8014.

**Table 1 microorganisms-12-00181-t001:** Basic genome information of test strains.

**Strain**	** *Lactiplantibacillus plantarum* **		
	**WLPL21**	**ZDY04**	**ATCC8014**
Genomic size	3,304,253	3,304,238	3,202,808
G + C content	44.56%	44.56%	44.43%
Number of ORFs	3072	3075	3029
Number of tRNA	65	65	66
Number of rRNA	16	16	13
Number of nc RNA	61	61	4
Number of repeat regions	3	3	0

**Table 2 microorganisms-12-00181-t002:** Distribution of transport genes in LAB.

Strain	*Lactiplantibacillus plantarum*	
	WLPL21/ZDY04	ATCC8014
Transport system genes	386	232
ABC transport system genes	191	137
Betaine/proline/choline family ABC transporter ATP-binding protein gene	1	1
Peptide ABC transport genes	19	7
Energy-coupling factor ABC transporter genes	6	4
Sugar ABC transporter genes	9	4
ABC transporter ATP-binding protein genes	71	49
Amino acid ABC transporter genes	24	13
Phosphotransferase system (PTS) genes	65	17

**Table 3 microorganisms-12-00181-t003:** Plantaricin (*pln*) gene list of test strains.

*pln* Genes	*Lactiplantibacillus plantarum*		
	WLPL21	ZDY04	ATCC8014
*pln*A	+	+	-
*pln*B	+	-	-
*pln*E	+	+	+
*pln*F	+	+	+
*pln*I	+	+	-
*pln*J	+	+	-
*pln*K	-	+	-
*pln*M	+	+	-
*pln*N	+	+	-
*pln*P	+	+	-
*pln*Q	+	+	-
*pln*S	+	-	-
*pln*U	+	-	-
*pln*V	+	-	-
*pln*W	+	-	-
*pln*Y	+	-	-

## Data Availability

The data used to support the findings of this study are available from the corresponding author upon request.

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
