# Peer review of "Genome Analysis for Cholesterol-Lowing Action and Bacteriocin Production of Lactiplantibacillus plantarum WLPL21 and ZDY04 from Traditional Chinese Fermented Foods"

_microorganisms, 2024, doi:10.3390/microorganisms12010181_

Round 1
Reviewer 1 Report
Comments and Suggestions for Authors
The work is devoted to an urgent problem related to the study of Lactiplantibacillus plantarum used in Traditional fermented food. It was shown that L. plantarum WLPL21 from fermented soybeans and ZDY04 15 from pickled vegetables could remove cholesterol 1.89 and was 1.61 times more active than the collection strain L. plantarum ATCC8014.
The work was carried out at a good methodological level, primarily from the point of view of bioinformation analysis. However, no detailed comparison (Figures 1-4, Table 2) was made after whole-genome sequencing of strains WLPL21 and ZDY04 with the previously known sequence ATCC8014. Moreover, for strains WLPL21 and ZDY04 the information in Figures 2-4 and Table 2 is duplicated; duplicates should be removed and signed that the information provided is valid for both strains (which must be compared with ATCC8014). The work is called “Genome analysis for cholesterol-lowing strains of Lactiplantibacillus plantarum WLPL21 and ZDY04 from traditional Chinese fermented foods,” but the authors did not directly find factors leading to cholesterol-lowing activity. However, gene clusters for bacteriocins production have been identified. It turned out that in LAB ecological strains (WLPL21 and ZDY04) the clusters of genes for bacteriocins production are significantly expanded compared to ATCC8014. This is one of the key results of the work, which should have been added to the title of the work along with “cholesterol-lowing strains.”
The work may be published after revision taking into account the comments made.
Comments
The abstract is 264 words long, well over the 200 word limit. Reformulate the abstract, leaving only key information, in order to more succinctly present the results obtained in a text volume of up to 200 words.
Lines 22-26
Reword the sentence. In the current version, it looks too cumbersome, since it provides information on 3 parameters in two strains, first in absolute units, then also relative to the L. plantarum ATCC8014 strain. Break this complex sentence into several simpler ones, or remove either the absolute or the relative units. Readers will be able to obtain more complete information in the Results section.
Lines 35-37
The review https://doi.org/10.3390/microorganisms10061151 on Traditional Fermented Foods reviews numerous studies going back to 2014. In this regard, check whether TFF and have attracted global attentions in the recent five years, and not 10 years or more
Lines 76-79
Indicate the source of strains L. plantarum WLPL21 and L. plantarum ZDY04. From what collection did they come from, or in what work were they isolated?
Line 75. 2.1. Bacterial strain and culture conditions
Add information about L. plantarum ATCC8014 to this section
Lines 81-82
Describe in more detail the method for isolating genomic DNA (what method was used, what kit was used for this) or refer to a literary source by analogy with which you carried out the extraction.
Figure 2-4
In Figures 2, 3 and 4 you have almost the same information, except for 640 and 641 unknown 161 genes in L. plantarum WLPL21 and ZDY04. In this regard, it seems uninformative to show the same thing twice. Combine Figures 2A and 2B; 3A and 3B; 4A and 4B. Apply existing differences. This will be much clearer for the reader, since it is uninformative to look through the same values, making sure that they coincide.
Also provide information for ATCC 8014 as you did in Table 3.
Figure 7
Expand the legend of Figure 7, write what is marked in yellow, which areas are highlighted in red, green and blue.
Lines 208-210
Expand the description of the results obtained as a result of Genome collinearity analysis. In the current version, the description of the results is only qualitative and not quantitative.
Lines 230 (Table 2. Distribution of transport genes in LAB)
Combine columns WLPL21 and ZDY04 because you have the same results for the two strains. It would also be more informative to compare these results with strain ATCC8014. Provide information for ATCC 8014 as you did in Table 3.
Line 277
Correct to L. plantarum
Comments on the Quality of English Language
The work is devoted to an urgent problem related to the study of Lactiplantibacillus plantarum used in Traditional fermented food. It was shown that L. plantarum WLPL21 from fermented soybeans and ZDY04 15 from pickled vegetables could remove cholesterol 1.89 and was 1.61 times more active than the collection strain L. plantarum ATCC8014.
The work was carried out at a good methodological level, primarily from the point of view of bioinformation analysis. However, no detailed comparison (Figures 1-4, Table 2) was made after whole-genome sequencing of strains WLPL21 and ZDY04 with the previously known sequence ATCC8014. Moreover, for strains WLPL21 and ZDY04 the information in Figures 2-4 and Table 2 is duplicated; duplicates should be removed and signed that the information provided is valid for both strains (which must be compared with ATCC8014). The work is called “Genome analysis for cholesterol-lowing strains of Lactiplantibacillus plantarum WLPL21 and ZDY04 from traditional Chinese fermented foods,” but the authors did not directly find factors leading to cholesterol-lowing activity. However, gene clusters for bacteriocins production have been identified. It turned out that in LAB ecological strains (WLPL21 and ZDY04) the clusters of genes for bacteriocins production are significantly expanded compared to ATCC8014. This is one of the key results of the work, which should have been added to the title of the work along with “cholesterol-lowing strains.”
The work may be published after revision taking into account the comments made.
Comments
The abstract is 264 words long, well over the 200 word limit. Reformulate the abstract, leaving only key information, in order to more succinctly present the results obtained in a text volume of up to 200 words.
Lines 22-26
Reword the sentence. In the current version, it looks too cumbersome, since it provides information on 3 parameters in two strains, first in absolute units, then also relative to the L. plantarum ATCC8014 strain. Break this complex sentence into several simpler ones, or remove either the absolute or the relative units. Readers will be able to obtain more complete information in the Results section.
Lines 35-37
The review https://doi.org/10.3390/microorganisms10061151 on Traditional Fermented Foods reviews numerous studies going back to 2014. In this regard, check whether TFF and have attracted global attentions in the recent five years, and not 10 years or more
Lines 76-79
Indicate the source of strains L. plantarum WLPL21 and L. plantarum ZDY04. From what collection did they come from, or in what work were they isolated?
Line 75. 2.1. Bacterial strain and culture conditions
Add information about L. plantarum ATCC8014 to this section
Lines 81-82
Describe in more detail the method for isolating genomic DNA (what method was used, what kit was used for this) or refer to a literary source by analogy with which you carried out the extraction.
Figure 2-4
In Figures 2, 3 and 4 you have almost the same information, except for 640 and 641 unknown 161 genes in L. plantarum WLPL21 and ZDY04. In this regard, it seems uninformative to show the same thing twice. Combine Figures 2A and 2B; 3A and 3B; 4A and 4B. Apply existing differences. This will be much clearer for the reader, since it is uninformative to look through the same values, making sure that they coincide.
Also provide information for ATCC 8014 as you did in Table 3.
Figure 7
Expand the legend of Figure 7, write what is marked in yellow, which areas are highlighted in red, green and blue.
Lines 208-210
Expand the description of the results obtained as a result of Genome collinearity analysis. In the current version, the description of the results is only qualitative and not quantitative.
Lines 230 (Table 2. Distribution of transport genes in LAB)
Combine columns WLPL21 and ZDY04 because you have the same results for the two strains. It would also be more informative to compare these results with strain ATCC8014. Provide information for ATCC 8014 as you did in Table 3.
Line 277
Correct to L. plantarum
Author Response
Dear editors and reviewers:
Thanks for your kindness advice for our manuscript (microorganisms-2805350) titled as “Genome analysis for cholesterol-lowing strains of Lactiplantibacillus plantarum WLPL21 and ZDY04 from traditional Chinese fermented foods”. Those comments are all valuable and very helpful for revising and improving it, as well as very important for guiding our future research. We have comprehensively considered your opinions and tried our best to revise the manuscript according to the Reviewers’ pertinent comments. We hope that the revision will meet with your requirements and are looking forward to your positive decision.
Yours sincerely
Dr. Hua Wei
State Key Laboratory of Food Science and Resources, Nanchang University, Nanchang 330047, P. R. China.
Tel: +86-791-88334578;
Fax: +86-791-8833-3708;
E-mail: weihua@ncu.edu.cn
Answers to Reviewer’s comments:
Reviewer 1:
Point 1: The work was carried out at a good methodological level, primarily from the point of view of bioinformation analysis. However, no detailed comparison (Figures 1-4, Table 2) was made after whole-genome sequencing of strains WLPL21 and ZDY04 with the previously known sequence ATCC8014. Moreover, for strains WLPL21 and ZDY04 the information in Figures 2-4 and Table 2 is duplicated; duplicates should be removed and signed that the information provided is valid for both strains (which must be compared with ATCC8014). The work is called “Genome analysis for cholesterol-lowing strains of Lactiplantibacillus plantarum WLPL21 and ZDY04 from traditional Chinese fermented foods,” but the authors did not directly find factors leading to cholesterol-lowing activity. However, gene clusters for bacteriocins production have been identified. It turned out that in LAB ecological strains (WLPL21 and ZDY04) the clusters of genes for bacteriocins production are significantly expanded compared to ATCC8014. This is one of the key results of the work, which should have been added to the title of the work along with “cholesterol-lowing strains.”
Response 1: Thank you for your suggestion. We have combined the origin figure 2a with 2b, figure 3a with 3b, and named them as new figure 2 and 3. Its legends were revised also in line 212-217.
As for comparison of L. plantarum ATCC8014, our opinion is as follows:
- Routinely, for genome analysis of our strains of circle genome map, COG, KEGG and GO, standard strains were not included [1-3]; 2. Due to limited space of this manuscript, the analysis of L. plantarum ATCC8014 was not included; 3. We have obtained several other excellent probiotics isolated from traditional Chinese fermented foods and will prepare genomic analysis and function proving test, we will include L. plantarum ATCC8014 as a control strain in our work, just as your keen suggestion.
- Boucard, A. S., Florent, I., Polack, B., Langella, P., Bermudez, L. G., Genome sequence and assessment of safety and potential probiotic traits of Lactobacillus johnsonii CNCM I-4884. Microorganisms, 2022, 10(2): p. 273. DOI: 10.3390/microorganisms10020273.
- Kim, H., Yoo, M. S., Jeon, H., Shim, J. J., Park, W. J., Kim, J. Y., Lee, J. L., Probiotic properties and safety evaluation of Lactobacillus plantarum HY7718 with superior storage stability isolated from fermented squid. Microorganisms, 2023, 11(9): p. 2254. DOI: 10.3390/microorganisms11092254.
- Garcia, N., Bottacini, F., Van, S. D., Gahan, C. G. M., & Corsetti, A., Comparative Genomics of Lactiplantibacillus plantarum: insights into probiotic markers in strains isolated from the human gastrointestinal tract and fermented foods. Micro., 2022. 13: p. 854266. DOI: 10.3389/fmicb.2022.854266.
In table 2, we have added the transport genes of L. plantarum ATCC8014 in line 276. Correspondingly, description of this result was added in line 266-269 and 274-275.
In addition, our title has been changed to “Genome analysis for cholesterol-lowing and bacteriocin production of Lactiplantibacillus plantarum WLPL21 and ZDY04 from traditional Chinese fermented foods” in line1-3.
Point 2: The abstract is 264 words long, well over the 200 words limit. Reformulate the abstract, leaving only key information, in order to more succinctly present the results obtained in a text volume of up to 200 words.
Response 2: Thank you for your suggestion. We renew our abstract much more concise, which was not over 200 words as shown in line 12-25.
Point 3: Lines 22-26
Reword the sentence. In the current version, it looks too cumbersome, since it provides information on 3 parameters in two strains, first in absolute units, then also relative to the L. plantarum ATCC8014 strain. Break this complex sentence into several simpler ones, or remove either the absolute or the relative units. Readers will be able to obtain more complete information in the Results section.
Response 3: Thank you for your suggestion. We have simplified this sentence in line 19-20. “The survival rate of cell counts of L. plantarum WLPL21 and ZDY04 in 1% H2O2, pH 3.0 and 0.3% bile salt were higher than that of L. plantarum ATCC8014.”
Point 4: Lines 35-37
The review https://doi.org/10.3390/microorganisms10061151 on Traditional Fermented Foods reviews numerous studies going back to 2014. In this regard, check whether TFF and have attracted global attentions in the recent five years, and not 10 years or more
Response 4: Thank you for your kind remind. We have checked some literatures in recent 5 years. Some references are listed:
Liang, Y., Wenlai, F., Yan, H., Metaproteomics insights into traditional fermented foods and beverages. Compr. Rev. Food Sci. F., 2020, 19(5), p. 2506-2529, DOI: 10.1111/1541-4337.12601.
Anal, A. K., Perpetuini, G., Petchkongkaew, A., et. al., Food safety risks in traditional fermented food from South-East Asia. Food Con., 2020, 109, p. 106922, DOI: 10.1016/j.foodcont.2019.106922.
Tamang, J. P., Lama, S., Probiotic properties of yeasts in traditional fermented foods and beverages. J. Appl. Micro., 2022, 132(5), p. 3533-3542, DOI: 10.1111/jam.15467.
Wang, J. X., Hao, S. Y., Ren, Q., Uncultured microorganisms and their functions in the fermentation systems of traditional Chinese fermented foods. Foods, 2023, 12(14), p. 2691, DOI: 10.3390/foods12142691.
Lenini, C., Anala, F. R., Goni, A. J., et. al. Probiotic properties of Bacillus subtilis DG101 isolated from the traditional Japanese fermented food natto. Front. Micro., 2023, 14, p. 1253480, DOI: 10.3389/fmicb.2023.1253480.
Point 5: Lines 76-79
Indicate the source of strains L. plantarum WLPL21 and L. plantarum ZDY04. From what collection did they come from, or in what work were they isolated?
Response 5: Thank you for your kind remind. In our previous work, L. plantarum WLPL21 and ZDY04 were isolated from Chinese fermented food soybeans and pickled vegetables, respectively, and reference was added in line 82.
Zhao, K., Qiu, L., He, Y., Tao, X. Y., Zhang, Z. H., & Wei, H., Alleviation syndrome of high-cholesterol-diet-induced hypercholesterolemia in mice by intervention with Lactiplantibacillus plantarum WLPL21 via regulation of cholesterol metabolism and transportation as well as gut microbiota. Nutrients, 2023. 15(11): p. 2600. DOI: 10.3390/nu15112600
Point 6: Line 75. 2.1. Bacterial strain and culture conditions
Add information about L. plantarum ATCC8014 to this section
Response 6: Thank you for your kind suggestion. Some culture information about L. plantarum ATCC8014 was descripted in line 139-141. “L. plantarum WLPL21, ZDY04 and ATCC8014 were cultured under anaerobic conditions at 37°C at 12 h in MRS broth.”
Point 7: Lines 81-82
Describe in more detail the method for isolating genomic DNA (what method was used, what kit was used for this) or refer to a literary source by analogy with which you carried out the extraction.
Response 7: Thank you for your suggestion. We had added the method for isolating genomic DNA in line 86-96.
“Genomic DNA of L. plantarum WLPL21 and ZDY04 was extracted and purified based on method of Surachat et. al. described previously with minor change [19]. DNA extraction was performed with DNeasy extraction kit (QIAGEN, Hilden, Germany) following the manufacturer’s instructions. Briefly, the bacterial cell pellet (109 cfu) was resuspended in 180 μL of enzymatic lysis buffer, and incubated at 37°C for 30 minutes. 25 μL of proteinase K 200 μl Buffer AL (NaOH) was then added and mixed before incubating at 56°C for 30 min. Then, 200 μL of ethanol were added to the DNA sample and centrifuged through the DNeasy Mini spin column at 610 g for 1 minute. After that, DNA was washed with 500 μL of Buffer AW2 (75% ethanol) and then eluted with buffer AE (NaCl). Finally, the DNA concentration in the eluate was measured by a spectrophotometer at 260 nm. The ratio of the absorbance at 260 nm and 280 nm (A260/A280) provided an estimate of the purity of DNA by agarose gel electrophoresis.”
- Surachat, K., Sangket, U., Deachamag, P., & Chotigeat, W., In silico analysis of protein toxin and bacteriocins from Lactobacillus paracasei SD1 genome and available online databases. Plos one, 2017, 12(8): p. e0183548. DOI: 10.1371/journal.pone.0183548
Point 8: Figure 2-4
In Figures 2, 3 and 4 you have almost the same information, except for 640 and 641 unknown 161 genes in L. plantarum WLPL21 and ZDY04. In this regard, it seems uninformative to show the same thing twice. Combine Figures 2A and 2B; 3A and 3B; 4A and 4B. Apply existing differences. This will be much clearer for the reader, since it is uninformative to look through the same values, making sure that they coincide.
Response 8: Thank you for your suggestion. We have combined the origin figure 2a with 2b, figure 3a with 3b, and named them as figure 2 and 3. Its legends was revised also in line 212-217.
Point 9: Also provide information for ATCC 8014 as you did in Table 3.
Response 9: Thank you for your kind remind. The information for L. plantarum ATCC8014 has already been listed in Table 3.
Point 10: Figure 7
Expand the legend of Figure 7, write what is marked in yellow, which areas are highlighted in red, green and blue.
Response 10: Thank you for your kind remind. Yellow area was gene rearrangement region. Red, green and blue area represented corresponding gene regions. And we have added in line 252-253.
Point 11: Lines 208-210
Expand the description of the results obtained as a result of Genome collinearity analysis. In the current version, the description of the results is only qualitative and not quantitative.
Response 11: Thank you for your kind suggestion. We have added some analysis in line 247-250. “However, gene rearrangement region and different genome length indicated L. plantarum WLPL21 and ZDY04 may had closer relationship than L. plantarum ATCC8014. Therefore, the subtle differences in genome distribution may be related to the source of strains.”
Point 12: Lines 230 (Table 2. Distribution of transport genes in LAB)
Response 12: Thank you for your kind suggestion. We have added in table 2.
Point 13: Combine columns WLPL21 and ZDY04 because you have the same results for the two strains. It would also be more informative to compare these results with strain ATCC8014. Provide information for ATCC 8014 as you did in Table 3.
Response 13: Thank you for your kind suggestion. We have revised in Table 2. And Table 3 has already included the information of L. plantarum ATCC8014.
Point 14: Line 277
Correct to L. plantarum
Response 14: Thank you for your kind suggestion. We have revised in Line 324.

Reviewer 2 Report
Comments and Suggestions for Authors
Lactiplantibacillus plantarum, known for bacteriocins in fermented foods, was studied for its cholesterol-lowering potential. Genome analysis of WLPL21 and ZDY04, compared to ATCC8014, revealed 384 transport system genes and excellent cholesterol transport activity. In vitro, WLPL21 and ZDY04 removed cholesterol 1.89 and 1.61 times more than ATCC8014. They also exhibited higher survival rates in challenging conditions. Both strains encoded complete bacteriocin gene clusters, emphasizing their probiotic potential. This study elucidates the importance of combining genomic analysis and in vitro evaluations to explore the functional capabilities of probiotics.
The manuscript is well written and the experiments are well performed.
Several points should be addressed.
Characters in figs are too small.
Table2,3 which strain is high in cholesterol transport should be described,
>
Figure 7. Genome collinearity analysis of L. plantarum WLPL21, ZDY04 and ATCC8014
A figure legend should be described.
P12L261
“The” ability of probiotics to tolerate…”the” survival….
articles of nouns are missing in many sentences.
Fig10 the reason why atcc8014 was sensitive to h2o2 should be discussed from genome data.
Comments on the Quality of English Language
articles of nouns are missing in many sentences.
Author Response
Dear editors and reviewers:
Thanks for your kindness advice for our manuscript (microorganisms-2805350) titled as “Genome analysis for cholesterol-lowing strains of Lactiplantibacillus plantarum WLPL21 and ZDY04 from traditional Chinese fermented foods”. Those comments are all valuable and very helpful for revising and improving it, as well as very important for guiding our future research. We have comprehensively considered your opinions and tried our best to revise the manuscript according to the Reviewers’ pertinent comments. We hope that the revision will meet with your requirements and are looking forward to your positive decision.
Yours sincerely
Dr. Hua Wei
State Key Laboratory of Food Science and Resources, Nanchang University, Nanchang 330047, P. R. China.
Tel: +86-791-88334578;
Fax: +86-791-8833-3708;
E-mail: weihua@ncu.edu.cn
Answers to Reviewer’s comments:
Reviewer 2:
Point 1: Characters in figs are too small.
Response 1: Thank you for your suggestion. We have magnified characters in figures.
Point 2: Table2,3 which strain is high in cholesterol transport should be described,
Response 2: Thank you for your suggestion. L. plantarum WLPL21 and ZDY04 had the same transport genes (386), which are much higher than that of L. plantarum ATCC8014 (232). “L. plantarum WLPL21 and ZDY04 possessed excellent transport activity compared with L. plantarum ATCC8014.” And we have described in line 273-275.
Point 3: Figure 7. Genome collinearity analysis of L. plantarum WLPL21, ZDY04 and ATCC8014
A figure legend should be described.
Response 3: Thank you for your suggestion. We have added figure legends in figure 7 and figure explaining in line 252-253.
Point 4: P12L261
“The” ability of probiotics to tolerate…”the” survival….
articles of nouns are missing in many sentences.
Response 4: Thank you for your suggestion. We have checked and added “the” in our article.
Point 5: Fig10 the reason why atcc8014 was sensitive to h2o2 should be discussed from genome data.
Response 5: Thank you for your suggestion. We have added it in line 373-374. “Consequently, the excellent antioxidant capacity of L. plantarum WLPL21 and ZDY04 may be associated with their source of Chinese traditional fermented foods.”

Round 2
Reviewer 1 Report
Comments and Suggestions for Authors
The authors responded to all the reviewer’s comments, took into account most of the comments, and made the necessary changes and additions. The work only benefited from this, making it look more holistic.